# From nicotine to SARS-CoV-2 antivirals with potent in vivo efficacy and a broad anti-coronavirus spectrum

Kaustav Khatua[1,11], Sandeep Atla[1,11], Demonta Coleman[1,11], Lauren R. Blankenship[1,11], Yugendar R. Alugubelli[1,11], Veerabhadra Vulupala [1,11], Xuejiao Shirley Guo[1], Hongjie Xia [2], Birte K. Kalveram[2], David H. Walker[3], Brett L. Hurst [4], Sathish Kumar[5], Chia-Chuan D. Cho[1], Shivangi Sharma [1], Kai Yang[1], Dorsa Rabie[1], Satyanarayana Nyalata[1], Benjamin W. Neuman [5] ✉, Xuping Xie [2,6] ✉, Shiqing Xu [1,7] ✉ & Wenshe Ray Liu [1,7,8,9,10] ✉

Anecdotal reports about smoking that might prevent SARS-CoV-2 infection inspire the search for nicotine and its pyrolysis products as inhibitors of the SARS-CoV-2 main protease (M^Pro). This effort leads to the discovery of 3-vinylpyridine as an M^Pro inhibitor. 3-Vinylpyridine resembles part of nirmatrelvir in binding to M^Pro but does not involve a critical interaction with residue E166, whose mutation has led to resistance to nirmatrelvir. Integration of the two molecules, followed by a medicinal chemistry campaign, produces several molecules with better in vitro potency than nirmatrelvir. Two lead molecules, YR-C-136 and SR-B-103, display better pharmacokinetic characteristics than nirmatrelvir in virus-challenged male mice and much better antiviral efficacy in virus-challenged female mice. Both molecules maintain high potency in inhibiting the nirmatrelvir-resistant M^Pro (E166V/L50F) variant. They also exhibit a broad and highly potent antiviral spectrum against most pathogenic coronaviruses. With high in vivo potency, both molecules are potentially standalone pan-antivirals for coronaviruses and may serve as countermeasures for future coronavirus outbreaks.

Coronavirus disease 2019 (COVID-19) is an infectious illness caused by severe acute respiratory syndrome coronavirus 2 (SARS-CoV-2)[1]. After emerging in late 2019, the disease rapidly spread across the globe, resulting in a pandemic declared in early 2020[2]. Despite significant progress in vaccination and drug treatment, SARS-CoV-2 continues to circulate and evolve, causing ongoing infections and sporadic localized outbreaks[3–5]. To date, the virus has caused nearly 800 million confirmed infections and over 7 million reported deaths worldwide[6]. SARS-CoV-2 is a positive-sense single-stranded RNA virus that depends on several essential enzymatic viral proteins to infect and replicate in human host cells[7,8]. Targeting these viral proteins has proven to be a logical and highly effective strategy for developing direct-acting antivirals[9]. Among these viral proteins, the main protease (M^Pro) has been a prominent drug target. M^Pro is a cysteine protease that cleaves two large viral polyproteins to produce the replicase complex required for virus genome replication and subgenomic RNA synthesis[10–12]. Due to its highly conserved sequence and structure across coronaviruses, inhibitors of SARS-CoV-2 M^Pro are likely to retain activity against other pathogenic coronaviruses, offering potential as pan-coronavirus antivirals[13]. Several M^Pro inhibitors have been approved for COVID-19 treatment, including nirmatrelvir (NMV) in combination with ritonavir by the U.S. FDA[14], ensitrelvir in Japan[15], and simnotrelvir as a combination therapy with ritonavir, atilotrelvir, and leritrelvir in China[16–18]. However, these therapies present certain limitations, including

potential drug-drug interactions, narrow treatment windows, modest reductions in symptom duration, and insufficient clinical data in some cases[19–22]. Additionally, widespread use has raised concerns about emerging drug-resistant mutations, such as the well-known E166V substitution in M[Pro], which diminishes the efficacy of nirmatrelvir[23,24]. As a result, the search for novel and potent M[Pro]-targeting antivirals that can overcome resistance to current treatments remains an urgent priority.

During the early stages of the COVID-19 pandemic, several observational studies and anecdotal reports drew attention to a seemingly paradoxical trend: a lower-than-expected prevalence of current smokers among hospitalized COVID-19 patients[25–27]. This observation led to speculation that smoking, or specifically nicotine, might confer some degree of protection against SARS-CoV-2 infection or disease progression. These early findings, despite being met with skepticism and contradictory with conclusions from later studies[28], spurred our interest in examining smoke-derived molecules as potential antiviral agents by targeting M[Pro]. We focused on nicotine and its pyrolysis products due to their rapid adsorption through the alveolar epithelium into the systemic circulation[29]. We previously crystallized the apo form of M[Pro] and employed a soaking strategy to investigate potential inhibitor binding[30]. This approach facilitated a quick fragment-based drug discovery process, allowing us to screen nicotine and its pyrolysis derivatives, including myosmine, 3-methylpyridine, 3-ethylpyridine, 3-vinylpyridine (VP), 3-allylpyridine, 3-pyridinecarbonitrile, β-nicotyrine, nornicotine, nicotyrine, anabasine, and cotinine by soaking apo M[Pro] crystals with each compound at a concentration of 100 mM[31]. The high concentration was utilized due to the small molecular size of these compounds. Among all tested compounds, only the VP-soaked crystals exhibited significant 2Fo-Fc electron density at the active site of M[Pro], shown in Fig. 1A, when VP was omitted during structure refinement. This electron density corresponded well with the structure of VP, enabling its conformational refinement. According to the Schechter-Berger nomenclature, the active site of M[Pro] comprises five subsites, S1, S2, S3, S4, and S1′, which interact with the P1, P2, P3, P4, and P1′ positions of peptide substrates, respectively[32–34]. The refined structure of the M[Pro]-VP complex revealed that VP binds neatly within

the S1 pocket, surrounded by residues F140, N142, the catalytic cysteine C145, H163, E166, and H172 (Fig. 1B and S12). The pyridine nitrogen atom of VP is in close proximity to the side-chain τ-nitrogen atom of H163, forming a hydrogen bond. Additionally, the vinyl group of VP adopts a conformation parallel to the phenyl side chain of F140, suggesting potential π–π stacking interactions. These interactions may explain why only VP was observed binding to the active site of M[Pro]. Using a previously established protocol[35], VP was characterized with an $IC_{50}$ value of 12 mM for M[Pro] inhibition (Table 1).

Developed by Pfizer, nirmatrelvir is an orally available antiviral that inhibits M[Pro][14]. It is co-administered with ritonavir, a CYP3A inhibitor that slows its metabolism, under the brand name Paxlovid. Nirmatrelvir is a tripeptidyl inhibitor featuring an activated C-terminal nitrile that covalently binds to the catalytic cysteine (C145) of M[Pro] to form a reversible thioimidate linkage. It also contains a P1 γ-lactam side chain that occupies the S1 subsite of M[Pro]. Structural superposition of the M[Pro]-nirmatrelvir complex (PDB entry: 7RFS) onto the M[Pro]-VP complex shown in Fig. 1C reveals that the γ-lactam moiety of nirmatrelvir aligns almost perfectly with the pyridine ring of VP. Specifically, the Cβ atom of nirmatrelvir's P1 residue overlaps precisely with the C5 atom of VP's pyridine ring, and the P1 Cα–Cβ bond is coplanar with and aligned to the C5–H bond of the pyridine ring. A hydrogen bond formed between the γ-lactam oxygen and the τ-nitrogen atom of M[Pro] H163 closely resembles the similar hydrogen bond formed between M[Pro] and the VP pyridine nitrogen atom. This close spatial match between the VP pyridine and the γ-lactam group of nirmatrelvir suggests a viable foundation for hybrid design. Integrating the chemical features of both molecules could guide the development of new M[Pro] inhibitors with improved potency, oral bioavailability, and pharmacological characteristics. There is also a notable distinction between the binding interactions of nirmatrelvir's P1 γ-lactam and VP within the M[Pro] S1 subsite. In nirmatrelvir, the nitrogen atom of the P1 γ-lactam forms a critical hydrogen bond with the side chain carboxylate of E166, a key interaction that enhances binding affinity. This hydrogen bond is a common feature among peptidyl inhibitors utilizing a P1 γ-lactam moiety, including simnotrelvir, atilotrelvir, and leritrelvir, as evidenced by their respective crystal structures shown in Fig. S13[16,36]. Ensitrelvir,

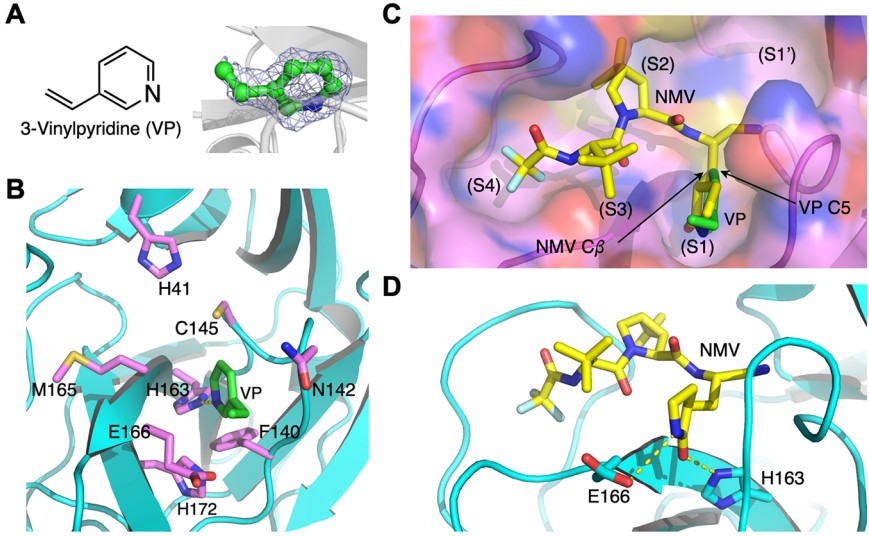

**Fig. 1 | 3-Vinylpyrine (VP) and its binding mode in the S1 pocket of SARS-CoV-2 M[Pro].** **A** The chemical structure of VP, along with its 2Fo-Fc electron density map contoured at the 1σ level in the active site of M[Pro]. **B** VP binding in the active site of M[Pro]. A hydrogen bond between the pyridine nitrogen of VP and an imidazole nitrogen of residue H163 is represented by a yellow dashed line. Surrounding side chains for residues in the S1 pocket are represented in sticks. **C**) Superimposition of the M[Pro]-VP complex over the M[Pro]-nirmatrelvir (NMV) complex (PDB entry: 7RFS)

around the active site. The five active site pockets, as defined by the Schechter-Berger nomenclature, are indicated in parentheses. The VP C-5 position and the Cβ atom in NMV connecting to the lactam side chain are labeled. The partially transparent contoured surface and secondary structure of M[Pro] are based on the pdb entry: 7RFS. **D** Two hydrogen bonds formed between nirmatrelvir's P1 lactam and two M[Pro] residues, H163 and E166 (PDB entry: 7RFS).

**Table 1 | Representative M^Pro inhibitors and their characteristics**

| ID[a] | Enzymatic IC$_{50}$ (μM) | Cellular IC$_{50}$ (μM) | Antiviral EC$_{50}$ (μM)[c] | CC$_{50}$ (μM) | Microsomal Stability t$_{1/2}$ (min) | Microsomal CL$_{int}$ (mL/min/kg) | PDB Code |
|---|---|---|---|---|---|---|---|
| VP | 11710 ± 962 | | | | | | 9BS7 |
| SR-A-37 | 87 ± 4.2 | | | | | | |
| SR-A-40 | 247 ± 24.5 | | | | | | |
| SR-B-51 | 0.044 ± 0.004 | 1.9 ± 0.09 | | | | | 9BTT |
| SR-B-77 | 0.10 ± 0.011 | 3.9 ± 0.33 | | | | | 9BTF |
| SR-B-78 | 0.038 ± 0.006 | 1.1 ± 0.16 | | | | | 9P6F |
| YR-C-108 | 0.043 ± 0.006 | 0.089 ± 0.009 | 0.031 ± 0.016 | 108 ± 19.6 | 82.5 ± 2.8 | 21 ± 0.7 | 9BTK |
| VB-D-142 | 0.054 ± 0.002 | 0.60 ± 0.028 | 0.068 ± 0.014 | 117 ± 7.3 | 54.7 ± 2.4 | 32 ± 1.4 | |
| SR-C-71 | 0.035 ± 0.004 | 0.067 ± 0.01 | 0.025 ± 0.009 | > 200 | 315 ± 19.5 | 5.5 ± 0.3 | |
| YR-C-155 | 0.075 ± 0.007 | 1.65 ± 0.4 | | | | | 9BVX |
| VB-C-153 | 0.035 ± 0.0003 | 0.21 ± 0.015 | 0.019 ± 0.004 | 131 ± 12.3 | 77 ± 5.6 | 22.7 ± 1.7 | |
| YR-C-136 | 0.035 ± 0.002 | 0.13 ± 0.008 | 0.023 ± 0.004 | 153 ± 23 | 140 ± 6 | 12.4 ± 0.6 | 9BSR |
| SR-B-103 | 0.030 ± 0.001 | 0.23 ± 0.011 | 0.045 ± 0.009 | 197 ± 6 | 150 ± 16 | 12 ± 1.2 | 9BVW |
| SR-B-122 | 0.045 ± 0.003 | 0.45 ± 0.018 | 0.036 ± 0.004 | 103 ± 12.5 | 75 ± 11 | 23.4 ± 3.3 | |
| VB-C-188 | 0.048 ± 0.001 | 0.51 ± 0.07 | 0.040 ± 0.008 | 174 ± 21 | 71 ± 9 | 24.6 ± 3.2 | |
| YR-C-120 | 0.037 ± 0.0005 | 0.69 ± 0.06 | 0.105 ± 0.034 | > 200 | 33.8 ± 0.6 | 51.3 ± 1 | |
| YR-D-51 | 0.15 ± 0.016 | 1.7 ± 0.23 | | | | | |
| YR-D-52 | 0.079 ± 0.005 | 0.83 ± 0.08 | 0.319 ± 0.176 | 104 ± 18 | 82.5 ± 1.4 | 21 ± 0.3 | |
| NMV | 0.066 ± 12[b] | 3.4 ± 0.8[b] | 0.044 ± 0.003 | 30.5 ± 5.5 | 173.6 ± 6.5 | 10 ± 0.4 | |

[a]All determined values are reported with two significant numbers. Reported values are given as the mean ± standard deviation of two individual experiments (n = 2). Source data are available in the Source Data file.
[b]Data were from ref. 39.
[c]In A549-hACE2 cells, against SARS-CoV-2 Delta variant.

despite being not a peptidyl inhibitor and lacking a γ-lactam, employs a 1-methyl-1*H*–1,2,4-triazol-3-yl group to bind the S1 subsite, which forms a hydrogen bond with the τ-nitrogen of H163. Additionally, its highly polarized 5-CH bond interacts with the carboxylate of E166, closely mimicking a hydrogen bond as shown in Fig. S2[37]. Consequently, all five clinically approved M^Pro inhibitors establish a hydrogen bond or hydrogen bond-like interaction with E166. In contrast, VP lacks such interaction with E166. Resistance studies have shown that mutations at E166, such as E166V, disrupt this critical hydrogen bond, leading to reduced efficacy of inhibitors like nirmatrelvir[23,38]. Since VP does not rely on this interaction with E166, VP-based M^Pro inhibitors may retain potency against SARS-CoV-2 strains harboring E166 mutations, positioning them as potential second-line therapeutics. Encouraged by the above-discussed observations, we initiated a medicinal chemistry campaign to develop hybrid molecules incorporating structural features from both VP and nirmatrelvir.

Here, we show that nicotine-derived fragment screening identifies 3-vinylpyridine as a ligand that binds the S1 subsite of SARS-CoV-2 M^Pro without engaging the resistance-prone residue E166. Guided by structural insights, we integrate features of VP with those of nirmatrelvir to develop a series of hybrid M^Pro inhibitors. Through iterative medicinal chemistry optimization, we identify two lead compounds, YR-C-136 and SR-B-103, that exhibit sub-100 nM enzymatic and cellular potency, favorable pharmacokinetic properties, and strong antiviral efficacy in virus-challenged mice. Both compounds retain high potency against the nirmatrelvir-resistant M^Pro (E166V/L50F) variant and display broad-spectrum antiviral activity against multiple SARS-CoV-2 variants and other pathogenic human coronaviruses.

## Results

### Structure-guided hybrid design yields potent M^Pro inhibitors

To simplify synthesis, we initially substituted VP with 3-chloropyridine in our designs (Fig. 2A). To replicate key M^Pro interactions observed in the P1 residue of nirmatrelvir, we attached an α-aminoacetonitrile moiety to the C5 position of 3-chloropyridine. The addition of an *N*-terminal acetamide cap yielded compound SR-A-37. SR-A-37 exhibited an IC$_{50}$ of 87 μM, a 138-fold improvement over VP, suggesting that the nitrile group likely recapitulates the covalent interaction with the catalytic cysteine of M^Pro. In most M^Pro-peptidic inhibitor complexes, the P1 α-amine forms a hydrogen bond with the backbone carbonyl oxygen of H164. To evaluate whether this interaction contributes to SR-A-37 binding, we synthesized SR-A-40, in which the α-amine was replaced with an oxygen atom, leading to the loss of this hydrogen bond. SR-A-40 showed an IC$_{50}$ value of 247 μM, nearly a 3-fold reduction in binding compared to SR-A-37, supporting the presence of a similar hydrogen bond in SR-A-37. Having established that SR-A-37 recaptures key interactions of the nirmatrelvir P1 residue, we incorporated the remaining structural features of nirmatrelvir to produce SR-B-51. SR-B-51 exhibited an IC$_{50}$ value of 44 nM, outperforming nirmatrelvir (66 nM) in the same assay[39]. The crystal structure of the M^Pro-SR-B-51 complex, obtained via soaking, revealed an electron density map at the active site that matched SR-B-51 precisely and showed covalent thioimidate bond formation with the active site cysteine (Fig. 2B). In the active site, SR-B-51 forms multiple hydrogen bonds. Its thioimidate nitrogen interacts with the backbone amide nitrogens of G143, S144, and C145, its P1 backbone nitrogen bonds with the carbonyl of H164, and its P2 backbone oxygen and nitrogen form two hydrogen bonds with the backbone nitrogen and oxygen, respectively,

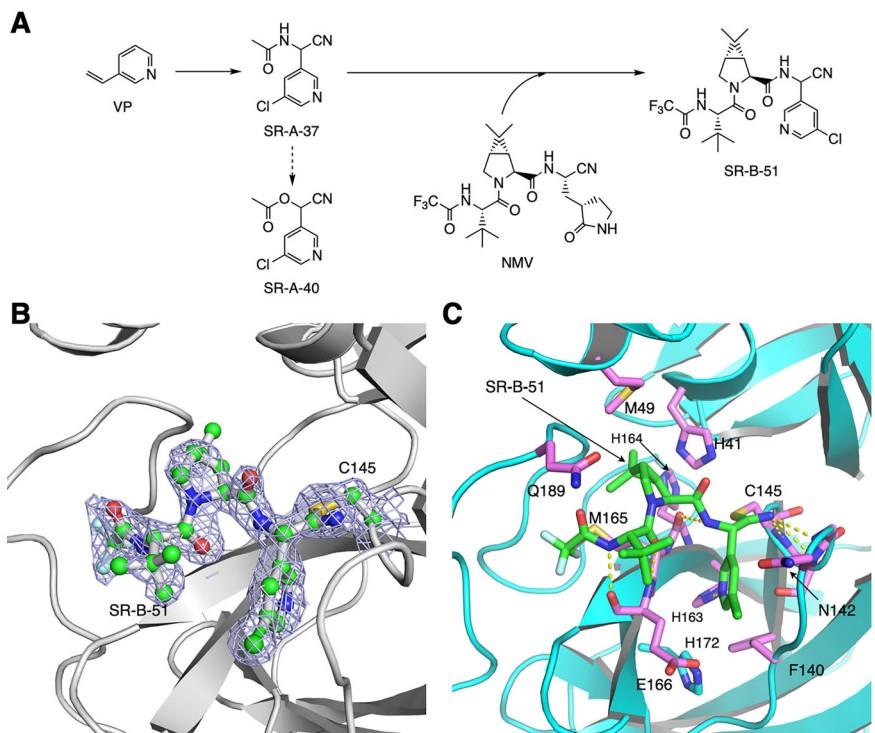

**Fig. 2 | From VP to SR-B-51. A** A rational design strategy leading to the development of SR-B-51. **B** 2Fo-Fc electron density map contoured at the 1σ level and surrounding SR-B-51 and residue C145 in the active site of M^Pro. SR-B-51 and C145 are covalently connected. **C** Atomic model of SR-B-51 bound at the active site of M^Pro. Residues surrounding SR-B-51 are presented in sticks. Hydrogen bonds between SR-B-51 and M^Pro are depicted as yellow dashed lines.

of E166. The *N*-terminal trifluoroacetamide cap occupies the S4 subsite through van der Waals interactions. Except at the S1 subsite, binding features for SR-B-51 largely mirror those of nirmatrelvir. In the S1 subsite, the chlorine atom in SR-B-51 displays well-defined electron density around it, allowing its position to be unambiguously refined. It is positioned to have van der Waals interactions with the side chains of residues N142 and E166. The side chain carboxylate of E166 has a distance longer than 3.5 Å to *m*-chloropyridine, ruling out any hydrogen bond-like close distance interactions with SR-B-51. The pyridine nitrogen of SR-B-51 forms a hydrogen bond with the side chain τ-nitrogen of H163. The slightly improved potency of SR-B-51 over nirmatrelvir may result from a better fit of *m*-chloropyridine within the S1 subsite than the lactam moiety in nirmatrelvir for van der Waals interactions (Fig. S14). Altogether, these findings confirm the success of our hybrid design strategy.

## Optimization of cellular and antiviral potency identifies lead compounds

When expressed in human cell lines such as HEK293T, M^Pro exhibits cytotoxic effects, leading to apoptosis. This cytotoxicity can be mitigated by cell-permeable M^Pro inhibitors, which rescue host cells. Leveraging this observation, we previously developed a cell-based assay to assess the intracellular potency of M^Pro inhibitors[40]. In this assay, HEK293T cells are transiently transfected to express an M^Pro-eGFP (enhanced green fluorescent protein) fusion protein and grown in the presence of an inhibitor. Effective inhibitors reduce M^Pro-mediated cytotoxicity, resulting in increased cell survival and enhanced expression of M^Pro-eGFP. This leads to the increase of fluorescence that can be quantitatively measured using flow cytometry, providing a readout of inhibitor efficacy within a cellular context. Notably, HEK293T cells express P-glycoprotein (P-gp), an efflux transporter known to reduce intracellular concentrations of certain drugs, including nirmatrelvir[41]. Using this assay, nirmatrelvir exhibited a determined cellular IC$_{50}$ value of 3.4 μM, approximately 50-fold higher

than its in vitro enzymatic IC$_{50}$ of 66 nM, highlighting the significant impact of P-gp-mediated efflux on cellular drug potency. While co-administration of ritonavir, a P-gp inhibitor, can enhance nirmatrelvir's intracellular concentration, as employed in the combination therapy Paxlovid, developing M^Pro inhibitors that function effectively as standalone agents offers advantages by minimizing potential drug-drug interactions. Therefore, we have been utilizing this cellular assay to prioritize M^Pro inhibitors with reduced susceptibility to P-gp efflux for further preclinical evaluation. To date, this assay has been employed to assess the intracellular potency of over 100 M^Pro inhibitors[39,42–45]. Applying this cell-based assay to SR-B-51 resulted in a cellular IC$_{50}$ value of 1.9 μM (Table 1), indicating improved intracellular potency compared to nirmatrelvir. However, this value remains substantially higher than SR-B-51's in vitro enzymatic IC$_{50}$ value, suggesting that factors such as cellular permeability or efflux mechanisms may limit its efficacy within cells. To address this discrepancy, we initiated a systematic structure-activity relationship (SAR) campaign aimed at optimizing SR-B-51's chemical structure to enhance its cellular potency and overall therapeutic potential.

Building upon prior findings that incorporating *O-tert*-butyl-threonine at the P3 position enhances the cellular and antiviral potency of peptidyl M^Pro inhibitors[30], we integrated this moiety into the SR-B-51 scaffold to synthesize YR-C-120 (Fig. 3A). To optimize interactions within the M^Pro S1 subsite, we explored various aromatic substitutions at the P1 position. This included *meta*-substituted pyridines, isoquinoline and its derivatives, mono- and difluoroisoquinolines, and 2,7- and 1,6-naphthyridines, resulting in the synthesis of 11 compounds. Among these, YR-C-136, featuring a 7-fluoroisoquinoline moiety at P1, demonstrated favorable pharmacokinetic properties and in vivo efficacy. Building on this, we retained the 7-fluoroisoquinoline at P1 and introduced alternative spirocyclic residues at the P2 position, specifically, (*S*)-2-azaspiro[4.4]nonane-3-carboxylate and (*S*)-1,4-dithia-7-azaspiro[4.4]nonane-8-carboxylate, to yield YR-D-51 and YR-D-52, respectively. The latter spiro residue is present in simnotrelvir. We

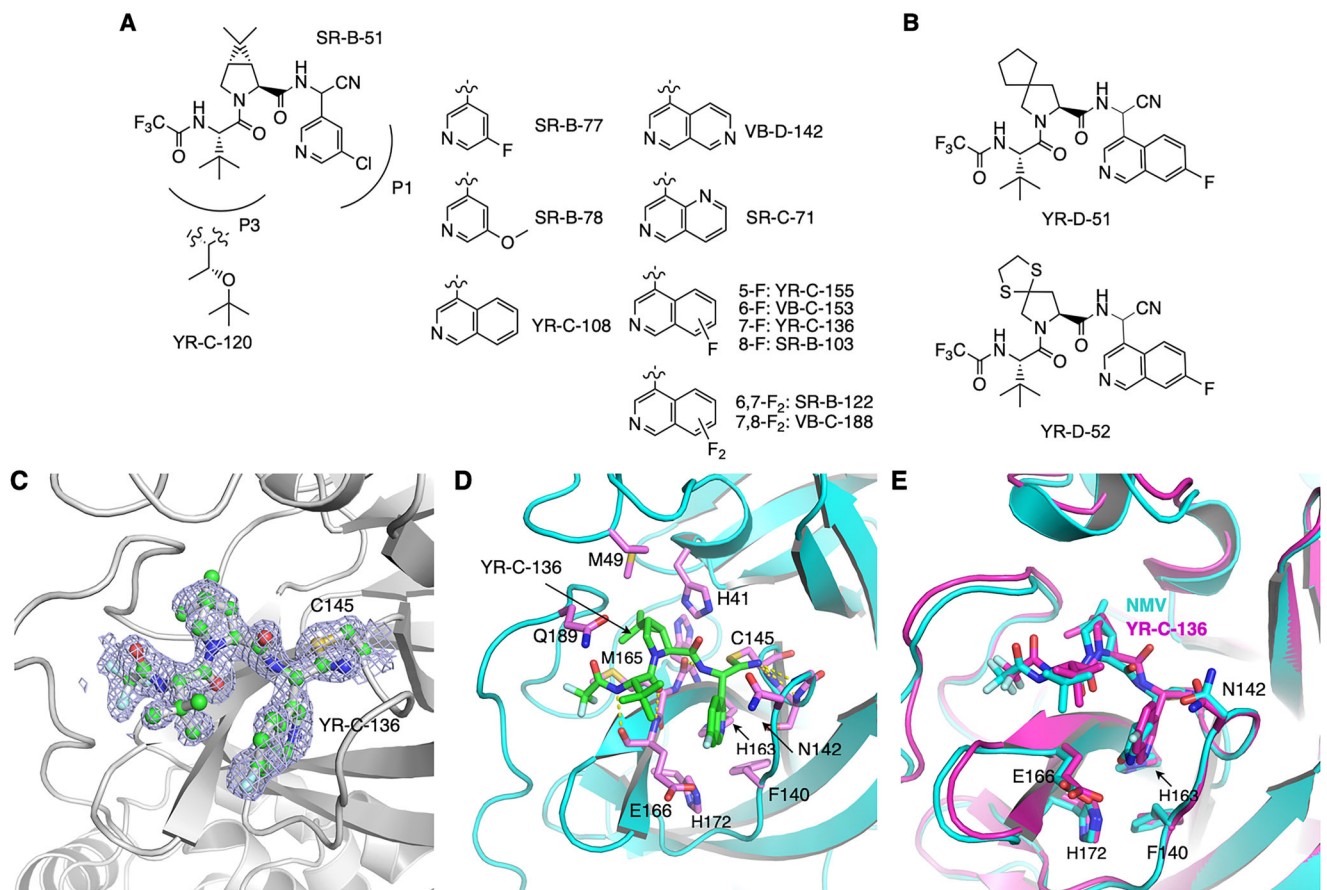

**Fig. 3 | Structure-activity relation exploration to identify more potent SARS-CoV-2 M^Pro inhibitors. A** Schematic diagram cd illustrating modifications at P1 and P3 sites of SR-B-51, leading to the generation of various inhibitors. **B** Two inhibitors based on YR-C-136 that have variations at the P2 site. **C** 2Fo-Fc electron density map contoured at the 1σ level, showing YR-C-136 and residue C145 in the active site of M^Pro. **D** Atomic model of YR-C-136 bound within the active site of M^Pro. Surrounding residues are shown as stick models. Hydrogen bonds between SR-B-51 and M^Pro are depicted as yellow dashed lines. **E** Superimposition of the M^Pro-YR-C-136 complex with the M^Pro-nirmatrelvir (NMV) complex (PDB entry: 7RFS), with residues surrounding the S1 subsite displayed in stick representation.

assessed the enzymatic inhibitory potency of these 14 compounds, with $IC_{50}$ values summarized in Table 1. Except for SR-B-77 and YR-D-51, which exhibited $IC_{50}$ values at or above 100 nM, all other inhibitors demonstrated $IC_{50}$ values below 100 nM. Six compounds, SR-B-78, SR-C-71, VB-C-153, YR-C-136, SR-B-103, and YR-C-120, exhibited $IC_{50}$ values below 40 nM. Among these, SR-B-103 showed the most potent enzymatic inhibition with an $IC_{50}$ value of 30 nM, more than twice as potent as nirmatrelvir, which has an $IC_{50}$ of 66 nM. These results demonstrate the effectiveness of strategic modifications mainly at the P1 position in enhancing the potency of M^Pro inhibitors.

Using the soaking strategy, we successfully determined the crystal structures of M^Pro in complexes with six newly developed inhibitors. Figure 3C shows a contoured 2Fo–Fc electron density map surrounding YR-C-136 at the active site of M^Pro, clearly confirming the formation of a covalent bond with the active site cysteine. As illustrated in Fig. 3D, YR-C-136 engages in all the hydrogen bonds previously observed with SR-B-51. YR-C-136 features a 7-fluoroisoquinoline moiety at the P1 position, which occupies the S1 subsite of M^Pro. Within this subsite, the nitrogen of the 7-fluoroisoquinoline ring forms a hydrogen bond with the τ-nitrogen of H163. Although this bicycloaromatic group is considerably larger than the m-chloropyridine in SR-B-51, the S1 subsite accommodates it without any steric clashes, as no surrounding residues are within unfavorable contact distance. As shown in Fig. 3E, superimposition with the M^Pro-nirmatrelvir complex reveals that most residues around the S1 subsite maintain similar conformations. The only notable difference is a slight reorientation of the side-chain amide

of N142, which adjusts to avoid steric hindrance with the 7-fluoroisoquinoline ring. Compared to the P1 γ-lactam of nirmatrelvir, which extends closer to the carboxylate of E166 to form a hydrogen bond, the 7-fluoroisoquinoline moiety of YR-C-136 makes tighter overall contact with the S1 subsite. Space-filled models shown in Fig. S15 further illustrate that the 7-fluoroisoquinoline ring achieves more extensive surface complementarity within the S1 subsite than the γ-lactam of nirmatrelvir, which may account for YR-C-136's nearly two-fold greater enzyme inhibition potency. Additionally, the fluorine atom, exposed to solvent and positioned within van der Waals distance of N142 and E166, may further enhance binding affinity.

The other five M^Pro-inhibitor complex structures determined include those for SR-B-77, SR-B-78, YR-C-108, YR-C-155, and SR-B-103. The overall binding mode of SR-B-77 closely resembles that of SR-B-51, with the key difference being a meta-fluoride substituent at the P1 pyridine ring. This small fluorine engages in less favorable van der Waals interactions with N142 and E166 (Fig. S16), which likely contributes to its reduced inhibitory potency. SR-B-78 features a methoxy group at the meta position of the P1 pyridine. This substituent introduces an additional hydrogen bond with the amide nitrogen of N142 (Fig. S17), providing a structural rationale for its enhanced potency relative to SR-B-51. Compared to YR-C-136, YR-C-108 lacks the 7-fluoro substituent on the isoquinoline ring but maintains a nearly identical binding mode in the M^Pro active site (Fig. S18). Although its potency is slightly lower than YR-C-136, this supports a modest but meaningful contribution of the 7-fluoro group to binding affinity for YR-C-136. YR-

C-155 represents a positional isomer of YR-C-136 in which the fluoro substituent is moved from the 7th to the 5th position of the isoquinoline ring. This design aimed to fill the spatial void between N142 and the P3 residue observed in YR-C-108 (Fig. S18B). The crystal structure of the $M^{pro}$-YR-C-155 complex confirms that the 5-fluoro group effectively occupies this space, forming van der Waals contacts with both N142 and the P3 residue in the inhibitor (Fig. S19). However, the inhibitor's binding to the $M^{Pro}$ active site reorients the P1 Cα−H bond to point directly toward the fluorine atom (Fig. S19C), a thermodynamically unfavorable arrangement. This steric penalty may explain why YR-C-155 exhibits lower potency than YR-C-108, despite the initial rationale for improved binding. In SR-B-103, the fluoro substituent is shifted from the 7th in YR-C-136 to the 8th position of the isoquinoline ring. This change was intended to bring the fluorine atom into closer proximity with the E166 side chain carboxylate. The crystal structure confirms that E166 maintains a conformation similar to that in the $M^{Pro}$-YR-C-136 complex (Fig. S20), and the fluorine atom lies just 2.7 Å from the E166 carboxylate. While this short distance would typically result in steric repulsion if no hydrogen bond were present, SR-B-103 exhibits even greater potency than YR-C-136. This suggests that a weak hydrogen bond, an O−H···F interaction, is formed, strengthening the overall interactions with $M^{Pro}$.

All newly developed inhibitors were evaluated for cellular potency using the cytoprotection assay based on $M^{Pro}$-induced toxicity in HEK293T cells. Except for SR-B-78, YR-C-155, and YR-D-51, all compounds exhibited cellular $IC_{50}$ values below 1 μM (Table 1), with two compounds achieving $IC_{50}$ values below 100 nM, indicating their less influence by the P-gp transporter compared to nirmatrelvir. To assess antiviral activity in a physiologically relevant system, we tested inhibitors with submicromolar cellular potency in A549-hACE2 cells, a well-established model for SARS-CoV-2 infection and antiviral evaluation. This cell line has minimal P-gp drug efflux, which decreases assay complications[46]. Using the SARS-CoV-2 strain USA_WA1/2020, we measured antiviral $EC_{50}$ values for all new compounds, which are summarized in Table 1. Except for YR-C-120 and YR-D-52, all compounds demonstrated antiviral $EC_{50}$ values below 100 nM. YR-C-120 contains an O-tert-butyl-threonine residue at the P3 position. Its relatively weak antiviral potency suggests that this moiety is not a favorable match with a P1 pyridine-based side chain. YR-D-52 differs from YR-C-136 only at the P2 position, containing (S)-1,4-dithia-7-azaspiro[4.4]nonane-8-carboxylate. Despite this single substitution, YR-D-52 shows a ~15-fold reduction in antiviral potency. It also exhibits weaker enzymatic and cellular potency compared to YR-C-136, indicating that the P2 (S)-1,4-dithia-7-azaspiro[4.4]nonane-8-carboxylate group is a suboptimal partner for a P1 pyridine scaffold. Antiviral potency was similar in Vero E6-ACE2-TMPRSS2 cells against eight SARS-CoV-2 strains, with antiviral $EC_{50}$ values of 15−97 nM for YR-C-136 and 31−153 nM for SR-B-103, compared to 1.5-5.7 μM for nirmatrelvir. All compounds were further assessed for cytotoxicity in HEK293T cells. As shown in Table 1, all inhibitors displayed lower cytotoxicity compared to nirmatrelvir, which has a $CC_{50}$ value of 31 μM. All inhibitors had $CC_{50}$ values above 100 μM, and two exceeded 200 μM. Selectivity indices ($CC_{50}/EC_{50}$) were calculated to assess therapeutic windows. Except for YR-D-52, all compounds had selectivity indices above 1000, with SR-C-71 displaying a selectivity index of 8000. These results collectively indicate that the newly developed inhibitors not only exhibit potent antiviral activity but also possess low cytotoxicity and high selectivity, representing a significant improvement over nirmatrelvir.

## Lead inhibitors exhibit in vivo efficacy and broad activity against resistant $M^{Pro}$ variants and pathogenic coronaviruses

In vitro metabolic stability parameters, including half-life ($t_{1/2}$) and intrinsic clearance ($Cl_{int}$) in human liver microsomes, were determined for all inhibitors exhibiting submicromolar cellular potency. The resulting data are summarized in Table 1. Compared to nirmatrelvir, SR-C-71 demonstrated improved metabolic stability, while YR-C-136 and SR-B-103 exhibited comparable profiles. Based on their favorable metabolic stability and overall performance, YR-C-136 and SR-B-103 were advanced for in vivo pharmacokinetic evaluation in C57BL/6 male mice, with nirmatrelvir included as a benchmark. SR-C-71 was not included in this evaluation due to its lethal effect at an intravenous (IV) injection dose of 20 mg/kg. YR-C-108 was included in the studies as a control from the group of compounds with a selectivity index of more than 3000, but in vitro half-life in human liver microsome below 100 minutes. Pharmacokinetic studies were conducted at 5 mg/kg for IV injection and 50 mg/kg for oral (PO) administration. Pharmacokinetic results, alongside nirmatrelvir, are shown in Fig. 4A, B. All three compounds exhibited slower clearance rates following IV administration. For oral dosing, YR-C-136 and SR-B-103 showed extended half-lives relative to nirmatrelvir and comparable $C_{max}$ levels (Table S1). In contrast, YR-C-108 had a significantly shorter half-life than nirmatrelvir, consistent with its in vitro half-life in human liver microsomes, potentially limiting its in vivo efficacy. YR-C-136 and SR-B-103 were subsequently advanced to in vivo efficacy evaluation. Eight- to ten-week-old BALB/c female mice were infected intranasally with a mouse-adapted SARS-CoV-2 strain[47]. Treatments were administered immediately after infection, followed by additional doses every 12 hours. At 48 hours post-infection, animals were euthanized for viral load and histopathological analyses. At an oral dose of 100 mg/kg, both compounds significantly reduced viral loads in the lungs of BALB/c female mice that were infected with a mouse-adapted SARS-CoV-2 strain when measured two days post-inoculation. Specifically, YR-C-136 and SR-B-103 lowered lung viral titers by approximately 120-fold and 74-fold, respectively, compared to the vehicle control. In contrast, nirmatrelvir at the same dose achieved only a 3-fold reduction in viral load (Fig. 4C). Histopathological analysis of lung tissue collected at 48 h post-infection (DPI) revealed that YR-C-136 treatment resulted in markedly fewer pulmonary lesions than vehicle treatment (Fig. 4D), indicating that YR-C-136 not only suppresses viral replication but also mitigates SARS-CoV-2-induced lung pathology.

The E166V/L50F mutation in $M^{Pro}$ has been shown to confer significant resistance to nirmatrelvir[48]. To evaluate the efficacy of alternative inhibitors, we expressed the mutant enzyme and tested its susceptibility to YR-C-136 and SR-B-103. As shown in Fig. 5A, both inhibitors retained high potency against the mutant protease, with $IC_{50}$ values of 117 nM for YR-C-136 and 75 nM for SR-B-103. These values represent only a 2−3-fold reduction in potency compared to their activity against the wild-type enzyme, indicating that both compounds remain effective despite the resistance mutation. In order to assess potentially generated resistance in the presence of SR-B-103 or YR-C-136, SARS-CoV-2 was grown in concentrations of SR-B-103 or YR-C-136 that were doubled each passage until the virus no longer produced cytopathic effects. Mutations resulting in amino acid changes to $M^{Pro}$ included T21I (7 of 8 replicates, 100% prevalence) and S144A (one SR-B-103 replicate, 100% prevalence), but not E166V or L50F. Both T21I and S144A are known from previous nirmatrelvir resistance studies, and S144A confers slower growth kinetics[49]. To assess cross-reactivity, we next expressed $M^{Pro}$ enzymes from SARS-CoV and MERS-CoV and evaluated the inhibition profiles of YR-C-136 and SR-B-103. The determined $IC_{50}$ values (Fig. 5B C) were below 500 nM for both proteases, indicating broad-spectrum inhibitory potential for the two inhibitors, albeit with reduced potency compared to SARS-CoV-2 $M^{Pro}$. Building on these findings, we further tested YR-C-136 and SR-B-103 against multiple SARS-CoV-2 variants, including USA_WA1/2020, Delta, and Omicron, and a panel of other pathogenic human coronaviruses: 229E (an alphacoronavirus), OC43 (a betacoronavirus), SARS-CoV, and MERS-CoV. Inhibition profiles across these strains are summarized in Fig. 5D−J. Both inhibitors exhibited $EC_{50}$ values below 100 nM against all three SARS-CoV-2 variants. For the currently dominant Omicron

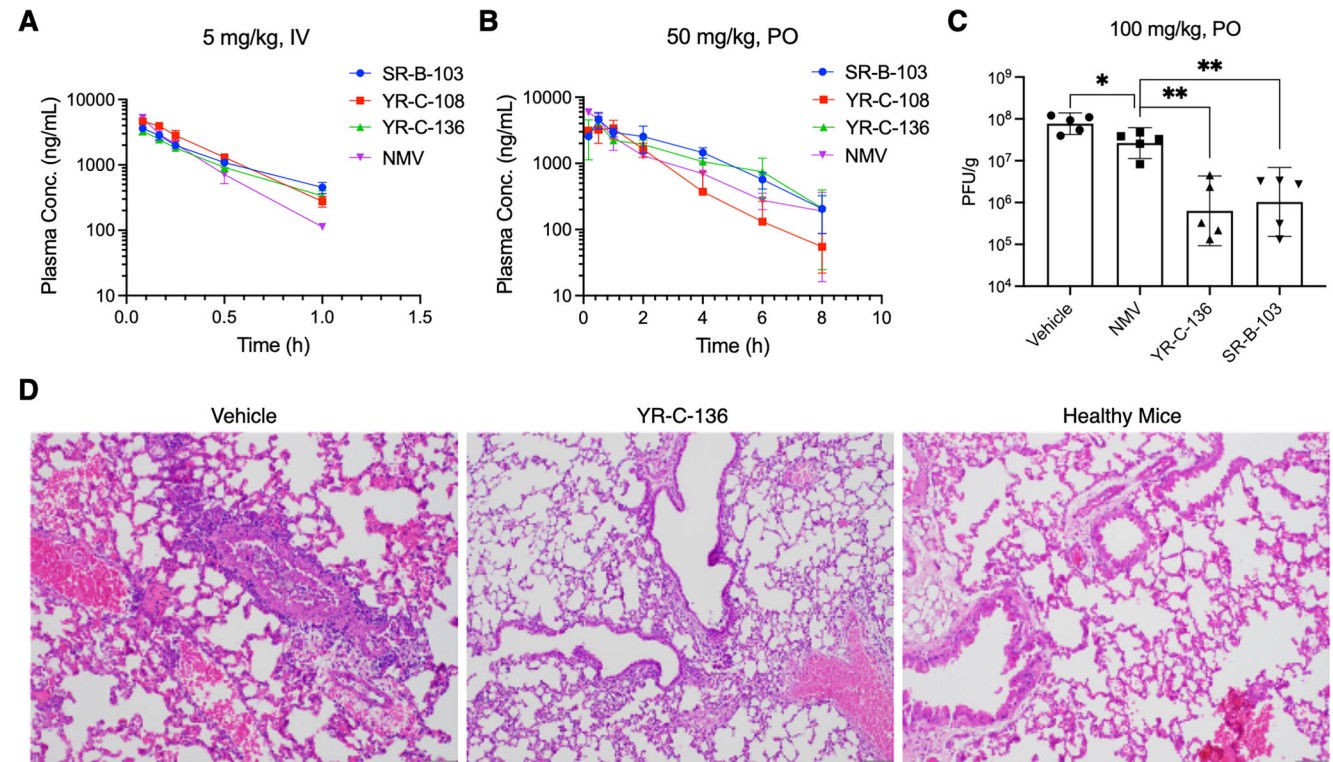

**Fig. 4 | In vivo PK profile and antiviral efficacy of selected M^Pro inhibitors.**
Plasma drug concentrations of SR-B-103, SR-B-108, and YR-C-136 in male C57BL/6J mice (aged 6–7 weeks) following (**A**) i.v. injection of 5 mg/kg and (**B**) p.o. administration of 50 mg/kg of indicated inhibitors in DMSO/PEG400/H$_2$O (10/30/60, v/v/v). NMV was used as a comparison. Values are reported as the mean of two independent experiments (*n* = 2). Source data are available in the Source Data file. **C** Viral titers in lungs collected at 48 h post-infection from vehicle or 100 mg/kg inhibitor-treated mice (*n* = 5 for each group). 8–10-week-old female BALB/c mice were infected with a mouse-adapted SARS-CoV-2 strain. Data are presented as geometric means ± 95% confidence intervals (CI). Statistical analyses were conducted using the two-tailed Mann-Whitney test. *$p < 0.05$; **$p < 0.01$. Source data are available in the Source Data file. **D** Lungs collected at 48 h post-infection from vehicle or 100 mg/kg YR-C-136-treated infected mice were stained with haematoxylin and eosin (H&E). Lungs from healthy mice were collected as well and used as a control. Scale bar: 50 μm.

variant, viral replication was reduced to less than 50% of control even at the lowest tested concentration (32 nM). For the four additional coronaviruses, both inhibitors showed EC$_{50}$ values below 1 μM. In particular, they displayed exceptional potency against OC43. Viral replication of OC43 was suppressed to less than 50% at 32 nM, and YR-C-136 completely inhibited OC43 at that concentration. Against MERS-CoV, both compounds exhibited EC$_{50}$ values around 500 nM, and slightly lower values were observed for SARS-CoV. These results clearly demonstrate that YR-C-136 and SR-B-103 possess broad-spectrum antiviral activity against diverse and clinically relevant coronaviruses. Their high potency, even in the presence of resistance mutations and across multiple viral species, highlights their potential as therapeutic candidates for current and future coronavirus outbreaks.

## Discussion

In summary, intrigued by anecdotal reports suggesting a potential protective effect of smoking against SARS-CoV-2 infection, we conducted a fragment-based screening of nicotine and its pyrolysis products as potential inhibitors for SARS-CoV-2 M^Pro. This effort led to the identification of VP, which binds to the S1 subsite of M^Pro without forming a hydrogen bond with the E166 residue, a feature that may reduce antiviral sensitivity to resistance mutations. Recognizing this potential, we integrated structural elements of VP with the clinically approved inhibitor nirmatrelvir and successfully developed a series of compounds with potent antiviral activity. Through iterative optimization, two lead candidates, YR-C-136 and SR-B-103, were identified based on their favorable pharmacokinetic profiles and strong in vivo antiviral efficacy. Both compounds retained high potency against the drug-resistant E166V/L50F M^Pro mutant and exhibited broad-spectrum antiviral activity across multiple SARS-CoV-2 variants and four additional human pathogenic coronaviruses. Importantly, YR-C-136 and SR-B-103 also demonstrated low susceptibility to P-gp efflux, distinguishing them from nirmatrelvir. Other non-covalent inhibitors, such as the Ugi-4-component reaction-derived ML188 and its analogs, have also provided important insights into SARS-CoV M^pro inhibition[50–52]. These pyridine-containing scaffolds, originally developed against SARS-CoV, were later shown to retain activity against SARS-CoV-2 M^Pro, highlighting their cross-reactive potential and utility in structure-based optimization. Incorporating such chemotypes into ongoing medicinal chemistry efforts may further expand the repertoire of tractable scaffolds for next-generation antiviral development. Taken together, these findings support the potential of YR-C-136 and SR-B-103 as standalone or second-line antiviral drug candidates for the treatment of SARS-CoV-2 and as countermeasures against future coronavirus outbreaks.

## Methods

### Expression of SARS-CoV-2 M^Pro, its E166V/L50F mutant, MERS-CoV M^Pro, and SARS-CoV M^Pro

**1.1 SARS-CoV-2 M^pro.** The expression plasmid pET28a-His-SUMO-CoV-2 M^Pro was constructed in a previous study (*35*). We used this construct to transform *E. coli* BL21(DE3) cells. A single colony grown on an LB plate containing 50 μg/mL kanamycin was picked and grown in 5 mL LB media supplemented with 50 μg/mL kanamycin overnight. We inoculated this overnight culture to 6 L 2YT media with 50 μg/mL kanamycin. Cells were grown to OD$_{600}$ as 0.8. At this point, we added

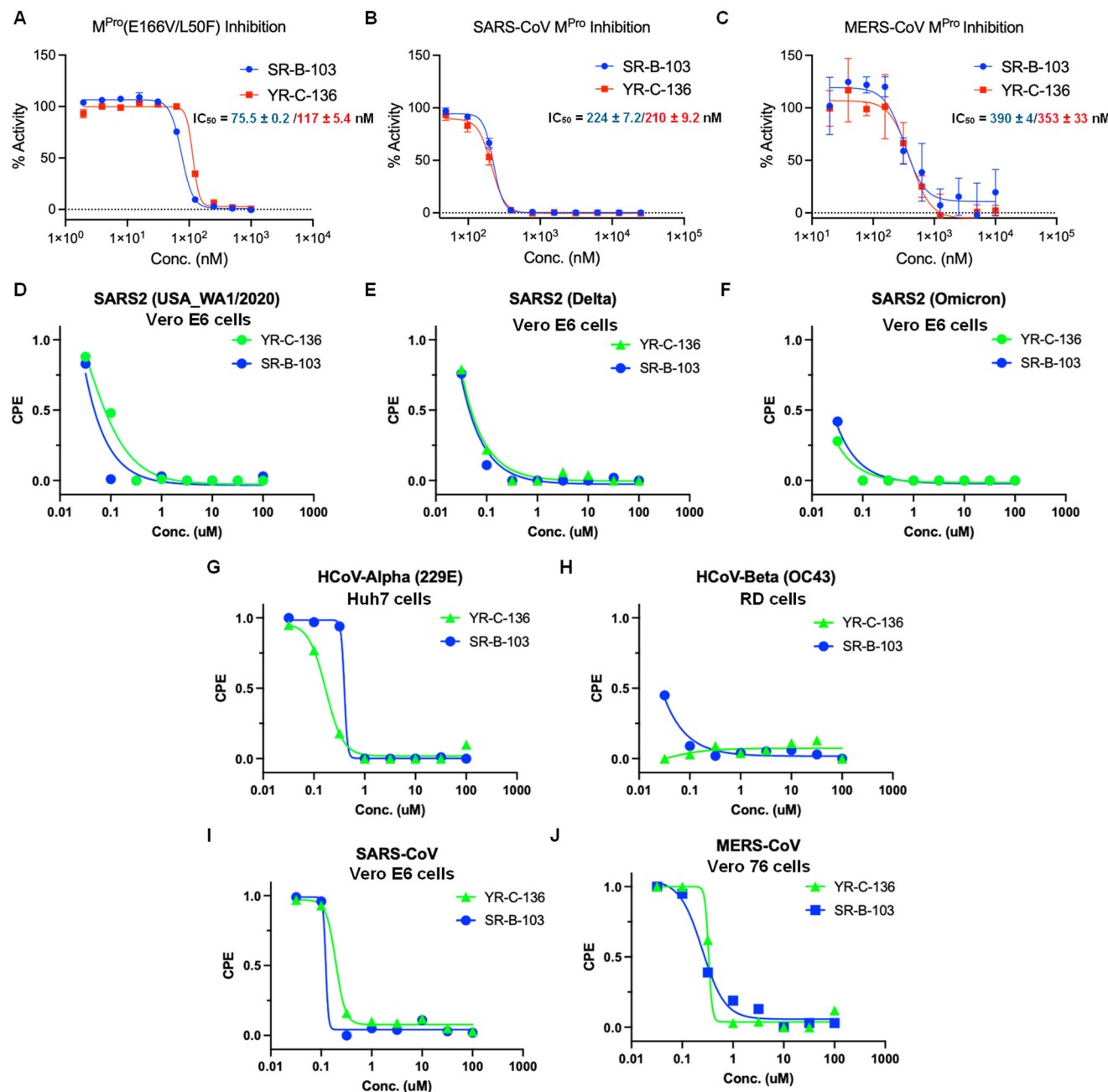

**Fig. 5 | Broad spectrum antiviral potency tests for YR-C-136 and SR-B-103.**
**A**–**C** Enzyme inhibition profiles of two compounds on SARS-CoV-2 M^Pro(E166V/L50F), SARS-CoV M^Pro, and MERS-CoV M^Pro. IC_{50} values are given as the mean ± standard deviation of two individual experiments ($n = 2$). Source data are available in the Source Data file. **D**–**J** their in vitro antiviral potency characterizations against three SARS-CoV-2 strains and four additional pathogenic coronavirus strains. Antiviral potency tests were based on the inhibition of cytopathogenic effect (CPE) induced by different viruses. Values are reported as the mean of two independent experiments ($n = 2$). Source data are available in the Source Data file.

1 mM IPTG to induce the expression of His-SUMO- CoV-2 M^Pro. Induced cells were let grown for 3 h and then harvested by centrifugation at 26934 × $g$, 4 °C for 30 min. We resuspended cell pellets in 150 mL lysis buffer (20 mM Tris-HCl, 100 mM NaCl, 10 mM imidazole, pH 8.0) and lysed the cells by sonication on ice. We clarified the lysate by centrifugation at 44155 × $g$, 4 °C for 30 min. We decanted the supernatant and mixed it with Ni-NTA resins (GenScript). We loaded the resins to a column, washed the resins with 10 volumes of lysis buffer, and eluted the bound protein using elution buffer (20 mM Tris-HCl, 100 mM NaCl, 250 mM imidazole, pH 8.0). We exchanged buffer of the elute to another buffer (20 mM Tris-HCl, 100 mM NaCl, 10 mM imidazole, 1 mM DTT, pH 8.0) using a HiPrep

26/10 desalting column (Cytiva) and digested the elute using 10 units SUMO protease overnight at 4 °C. The digested elute was subjected to Ni-NTA resins in a column to remove His-tagged SUMO protease, His-tagged SUMO tag, and undigested His-SUMO- CoV-2 M^Pro. We loaded the flow-through onto a Q-Sepharose column and purified CoV-2 M^Pro using FPLC by running a linear gradient from 0 to 500 mM NaCl in a buffer (20 mM Tris-HCl, 1 mM DTT, pH 8.0). Fractions eluted from the Q-Sepharose column were concentrated and loaded onto a HiPrep 16/60 Sephacryl S-100 HR column and purified using a buffer containing 20 mM Tris-HCl, 100 mM NaCl, 1 mM DTT, and 1 mM EDTA at pH 7.8. The final purified protein was concentrated and stored in a −80 °C freezer.

**1.2 SARS-CoV-2 M$^{pro}$ mutants (E166V/ L5OF).** To express M$^{pro}$(E166V/ L5OF), two mutations were introduced into pET28a-His-SUMO-M$^{pro}$ using two primer pairs for site-directed mutagenesis: E166V forward primer TGCACCATATGGTGTTACCGACCGGCGTACATGCC and reverse primer CGGTCGGTAACACCATATGGTGCATATAGCAAAAAGAG; L5OF forward primer CGAAGACATGTTTAACCCCAATTACGAGGATTTGCTTA TC and reverse primer TCGTAATTGGGGTTAAACATGTCTTCGGAG GTGCAGATGA. Expression, and purification, and processing M$^{pro}$ (E166V/L5OF) were the same as the wild type M$^{pro}$.

**1.3 MERS-CoV M$^{pro}$.** The MERS-M$^{pro}$ was identified from the NCBI database. DNAs for these genes were ordered from IDT DNA Inc. A partial SUMO C-terminus sequence was fused to the N-terminus of each sequence and purchased from IDT for cloning. Each gene was PCR amplified by 3Cpro-f (ACTCCTTAAGATTCTTGTACGACGG) and 3Cpro-r (GTGGTGGTGGTGCTCGAGTTA) and digested with AflII and XhoI, and gel extracted for later cloning. The pET28a-SUMO-M$^{pro}$ was digested with AflII and XhoI and ligated to each fragment by T4DNA ligase. Top10 chemical competent cells were transformed and sequence verified. Following sequence confirmation, the plasmid was used to transform *E. coli* BL21 cells. A single colony from an LB-kanamycin (50 µg/mL) plate was inoculated into 10 mL of 2YT medium with kanamycin and grown overnight at 37°C. The overnight culture was diluted 1:100 into 1 L of fresh 2YT medium containing 50 µg/mL kanamycin and incubated at 37°C until OD$_{600}$ reached ~0.8. Protein expression was induced by adding 1 mM IPTG, followed by overnight incubation at 16°C to enhance protein solubility. Cells were harvested by centrifugation at 5251 × $g$ for 10 min at 4 °C and stored at −80 °C until purification. Cell pellets were resuspended in 30 mL MERS-specific lysis buffer (20 mM Tris-HCl, 250 mM NaCl, 5% glycerol, 0.2% Triton X-100, 1 mM DTT, pH 8.5, supplemented with 1 mg/mL lysozyme). Cells were lysed by sonication on an ice water bath (65% amplitude, 1 sec on/4 sec off, total on-time 5 min), and the lysate was clarified by centrifugation at 44155 × $g$ for 30 min at 4 °C. The supernatant was applied to 6 mL Ni-NTA resin equilibrated with the MERS-specific lysis buffer. The flowthrough was reapplied to ensure complete binding. The resin was washed with 180 mL of the same buffer without 1 mg/mL lysozyme to remove non-specifically bound proteins. Bound MERS-M$^{pro}$ was eluted using elution buffer (20 mM Tris-HCl, 100 mM NaCl, 250 mM imidazole, 1 mM DTT, pH 8.0). The eluted protein was incubated overnight at 4 °C with SUMO protease (10 units/ mg) to remove the SUMO tag. The digest was buffer-exchanged into standard lysis buffer using a HiPrep 26/10 desalting column, and the desalted sample was applied to a second Ni-NTA column to remove His-tagged SUMO, SUMO protease, and any uncleaved fusion protein. The flow-through, containing cleaved MERS-M$^{pro}$, was collected. This sample was concentrated using a 10 kDa cutoff Amicon filter and subjected to size-exclusion chromatography on a HiLoad 16/600 Superdex 75 pg column equilibrated with 20 mM Tris-HCl, 100 mM NaCl, 1 mM DTT, 1 mM EDTA, pH 8.0. Fractions were analyzed by SDS-PAGE. Desired fractions were pooled, concentrated, quantified via A$_{660}$ assay, aliquoted, flash-frozen in liquid nitrogen, and stored at −80 °C.

**1.4 SARS-CoV M$^{pro}$.** The pET28a-SUMO-SARSM$^{pro}$ was purchased from Twist Bioscience. This plasmid was used to transform *E. coli* BL21 competent cells. A single colony from an LB-kanamycin (50 µg/mL) plate was inoculated into 10 mL 2YT medium containing 50 µg/mL kanamycin and grown overnight at 37 °C. The overnight culture was diluted 1:100 into 1 L of fresh 2YT medium with kanamycin and grown at 37 °C until the OD$_{600}$ reached ~0.8. Protein expression was induced by adding 1 mM IPTG, and the culture was incubated at 37 °C for 4 h. Cells were harvested by centrifugation at 5251 × $g$ for 10 min at 4 °C and stored at −80 °C. For purification, cell pellets were resuspended in 30 mL lysis buffer (20 mM Tris-HCl, 100 mM NaCl, 30 mM imidazole,

1 mM DTT, pH 8.0) containing 1 mg/mL lysozyme. Cells were lysed by sonication on ice (65% amplitude, 1 sec on/4 sec off, total on-time 5 min), and the lysate was clarified by centrifugation at 44155 × $g$ for 30 min at 4 °C. The supernatant was applied to 6 mL Ni-NTA resin pre-equilibrated with lysis buffer. The flow-through was reloaded to maximize binding. The column was washed with 180 mL of the lysis buffer and bound protein was eluted using the elution buffer (20 mM Tris-HCl, 100 mM NaCl, 250 mM imidazole, 1 mM DTT, pH 8.0). The eluted protein was incubated overnight at 4°C with SUMO protease (10 units/ mg) to cleave the SUMO tag. The digested sample was buffer exchanged into lysis buffer using a HiPrep 26/10 desalting column. The desalted protein was reloaded onto Ni-NTA resin to remove His-tagged SUMO, SUMO protease, and uncleaved fusion protein. The flow-through, containing cleaved SARS-M$^{pro}$, was collected. This flow-through was concentrated using a 10 kDa cutoff Amicon Ultra centrifugal filter and further purified by size-exclusion chromatography on a HiLoad 16/600 Superdex 75 pg column equilibrated with 20 mM Tris-HCl, 100 mM NaCl, 1 mM DTT, 1 mM EDTA, pH 8.0. Protein-containing fractions were identified by SDS-PAGE, pooled, concentrated, quantified using the A$_{660}$ assay, aliquoted, flash-frozen in liquid nitrogen, and stored at −80 °C.

### M$^{pro}$ inhibition characterizations for all synthesized inhibitors
For all inhibitors, we conducted the assay using 20 nM M$^{pro}$ and 10 µM Sub3 (DABCYL-Lys-Thr-Ser-Ala-Val-Leu-Gln-Ser-Gly-Phe-Arg-Lys-Met-Glu-EDANS). We dissolved all inhibitors in DMSO as 10 mM stock solutions. Sub3 was dissolved in DMSO as a 1 mM stock solution and diluted 100 times in the final assay buffer containing 10 mM Na$_x$H$_y$PO$_4$, 10 mM NaCl, 0.5 mM EDTA, and 1.25% DMSO at pH 7.6. We incubated M$^{pro}$ and an inhibitor in the final assay buffer for 30 min before adding the substrate to initiate the reaction catalyzed by M$^{pro}$. The production format was monitored in a fluorescence plate reader with excitation at 336 nm and emission at 490 nm. We calculated the initial rate according to the fluorescent intensity in the first 5 min by linear regression, which was then normalized according to the initial rate of positive and negative controls. We calculated the initial rate according to the fluorescent intensity in the first 5 min by linear regression, which was then normalized according to the initial rate of positive and negative controls.

### Cellular potency tests using HEK293T cells transiently expressing M$^{pro}$-eGFP
HEK 293T/17 cells were maintained in high-glucose DMEM supplemented with GlutaMAX and 10% fetal bovine serum in 10 cm culture plates at 37 °C and 5% CO$_2$ until reaching 80%–90% confluency. Cells were transfected with the pLVX-M$^{pro}$-eGFP-2 plasmid (Cao et al., 2022) using polyethyleneimine (30 µg/mL) and 8 µg of plasmid DNA in 500 µL opti-MEM per transfection. The transfection mixture was incubated with the cells overnight.

On the following day, the medium was removed, and cells were washed with PBS before enzymatic detachment using 0.05% trypsin-EDTA. The dissociated cells were resuspended in their original growth medium, and the density was adjusted to 5 × 10$^5$ cells/mL. A volume of 500 µL of the cell suspension was seeded into each well of a 48-well plate, followed by the addition of 100 µL of drug solution prepared in growth media. Cells were incubated under the same culture conditions for 72 hours with varying concentrations of inhibitors before flow cytometry analysis.

Post-incubation, cells were resuspended in 500 µL PBS and centrifuged at 800 × $g$ for 5 minutes. The supernatant was discarded, and cell pellets were resuspended in 200 µL PBS. eGFP fluorescence was analyzed using a Cytoflex Beckman Flow Cytometer, with cells sorted based on side scatter (SSC-A, SSC-H) and forward scatter (FSC-A). Gating was performed sequentially using SSC-A/FSC-A followed by SSC-A/SSC-H. eGFP fluorescence was excited using a 488 nm blue laser, and emissions were recorded at FITC-A (525 nm).

All processed data were plotted and fitted to a four-parameter Hill equation using GraphPad Prism 9.0 to determine $EC_{50}$ values.

## In vitro antiviral potency tests for SARS-CoV-2 and resistance profiling tests

Vero E6-ACE2-TMPRSS2 cells (BEI Resources, NR54970) and A549-hACE2 cells (BEI Resources, NR53821) were cultured in a 37 °C incubator with 5% $CO_2$ in DMEM supplemented with 10% fetal bovine serum, 1× antibiotic/antimycotic and puromycin dihydrochloride (10 μg/ml final concentration; added to maintain the transgenes in Vero cells only). SARS-CoV-2 strains WA1 (BEI Resources; USA-WA1/2020; NR52281), Alpha (BEI Resources; hCoV-19/USA/OR-OHSU-PHL00037/2021; NR55461), Beta (BEI Resources; hCoV-19/USA/MD-HP01542/2021; NR55282), Delta (BEI Resources; hCoV-19/USA/MD-HP20874/2021; NR56461), BQ.1.1 (local isolate; GISAID accession EPI_ISL_17371329), XBB.1.5 (BEI Resources; hCoV-19/USA/MD-HP40900/2022), BF.7 (BEI Resources; hCoV-19/USA/MD-HP38288/2022; NR58974) and CH.1.1 (local isolate; EPI_ISL_16702883) were cultured on Vero E6-ACE2-TMPRSS2. Infectious titer was determined by tissue culture infectious dose 50% ($TCID_{50}$) on Vero E6-ACE2-TMPRSS2 cells. Cells were inoculated at a multiplicity of 0.1 $TCID_{50}$ units per cell, incubated for 1 h at 37 °C to allow for virus adsorption, rinsed three times with phosphate-buffered saline pH 7 to remove unbound inoculum, and treated by the addition of DMEM with 10% fetal bovine serum and 1× antibiotic/antimycotic containing various concentrations of potential antivirals. Small aliquots of medium were collected at 48 h and 72 h after inoculation, and the amount of virus growth was titrated using a Luna One-Step RT175 qPCR (NEB, Ipswich, MA, US) with the N1 primer set targeting the nucleoprotein gene of the virus, calibrated to a standard curve of inactivated viral RNA (BEI Resources, NR-52285).

Resistance profiling was carried out using the BQ.1.1 strain of SARS-CoV-2, in four replicate cultures per antiviral, by serial passage of virus in concentrations of virus starting at the $EC_{50}$ value, and doubling in each subsequent virus passage, with each passage continuing for up to 96 h, until viral cytopathic effects (CPE), including syncytium formation and cell rounding were observed. Potentially resistant virus supernatant was collected from the passage with the highest concentration of each antiviral, where the virus was still able to produce CPE. Viral RNA was purified and sent to the Texas A&M University Institute for Genome Sciences and Society Molecular Genomics Core for whole-genome sequencing. For each sample, with library preparation using the xGen SARS-CoV-2 Amplicon Panel (Integrated DNA Technologies, https://www.idtdna.com). Sequencing was performed in an Illumina NovaSeq SP PE 2 × 150 flowcell version 1.5 (https://www.illumina.com) to generate an average of 3 million reads per sample.

## In Vitro Metabolic Stability Tests in Human Liver Microsomes

The metabolic stability measurements were based on previous publications and modified as described below. We determined the metabolic stability profile of the inhibitor, including $CL_{int, pred}$, and in vitro $t_{1/2}$ by the estimation of the remaining compound concentration after incubation with human liver microsome, NADPH (cofactor), EDTA, and $MgCl_2$ in a 0.1 M phosphate buffer (pH 7.4). We preincubated 5 μM of each inhibitor with 40 μL of human liver microsome (0.5 mg/mL) in 0.1 M phosphate buffer (pH 7.4) at 37 °C for 5 min. After preincubation, we added NADPH (5 mM, 10 μL) or 0.1 M PB (10 μL) to initiate the metabolic reaction at 37 °C. The reactions were conducted in triplicate. At 0, 5, 15, 30, 45, 60 min, we added 200 μL acetonitrile (with internal standard Diclofenac, 10 μg/mL) to quench the metabolic reaction. We then subjected the samples to centrifugation at 4 °C for 20 min at 3724 × g. Then we analyzed 50 μL of clear supernatants using LC-MS. We determined the percentage of remaining test compound using the formula: % remaining = (Area at $t_x$ /Average area at $t_0$) × 100. We calculated the half-life ($t_{1/2}$) using the slope (k) of the log-linear

regression of the curve for the % remaining parent compound versus time (min): $t_{1/2}$ (min) = $-\ln 2/k$. We determined $CL_{int, pred}$ (mL/min/kg) using the formula $CL_{int, pred} = (0.693/t_{1/2}) \times (1/$ (microsomal protein concentration (0.5 mg/mL)) × Scaling Factor (1254.16 for human liver microsome).

## In vivo pharmacokinetic characterizations for YR-C-136, SR-B-103, and YR-C-108

Male C57BL/6 mice, aged 6-7 weeks, were provided by BioLASCO Taiwan (under Charles River Laboratory Licensee). Space allocation for animals was 39 × 20 × 16 cm. All animals were maintained in a well-controlled temperature (20 - 24 °C) and humidity (30–70%) environment with 12 hours light/dark cycles. Free access to standard lab diet [MFG (Oriental Yeast Co., Ltd., Japan)] and autoclaved tap water was granted. All aspects of this work, including housing, experimentation, and animal disposal, were performed in general accordance with the "Guide for the Care and Use of Laboratory Animals: Eighth Edition" (National Academies Press, Washington, D.C., 2011) in the AAALAC-accredited laboratory animal facility. In addition, the animal care and use protocol was reviewed and approved by the IACUC at Pharmacology Discovery Services Taiwan (PDS), Ltd.

YR-C-136, SR-B-103, and YR-C-108 were formulated in dimethyl sulfoxide (DMSO)/polyethylene glycol 400 (PEG400)/ deionized water (WFI) (10/30/60, v/v/v) at 5 and 1 mg/mL with dosing volume at 10 and 5 mL/kg for each PO and IV administration, respectively. For each study ($n = 3$), YR-C-136, SR-B-103, and YR-C-108 were dosed iv (5 mg/kg) or via po gavage (50 mg/kg). Blood aliquots (-300 μL) were collected via cardiac puncture from mice into tubes coated with $K_2$EDTA at the specified time points. The tubes were mixed gently and kept on ice and then centrifuged at 2500 × g for 15 minutes at 4 °C, within 1 hour after collection. The plasma samples were processed using acetonitrile precipitation and analyzed by LC-MS/MS. The exposure levels (ng/mL) of YR-C-136, SR-B-103, or YR-C-108 in plasma samples were determined. The plot of plasma concentrations versus (vs.) time (mean ± SD) for YR-C-136, SR-B-103, or YR-C-108 was constructed. The fundamental PK parameters of YR-C-136, SR-B-103, or YR-C-108 after each PO ($t_{1/2}$, $T_{max}$, $C_{max}$, $AUC_{last}$, $AUC_{Inf}$, AUC/D, AUC Extr, MRT, Vz/F, CL/F) or IV ($t_{1/2}$, $T_{max}$, $C_{max}$, $AUC_{last}$, $AUC_{Inf}$, AUC/D, AUC Extr, MRT, Vz/F, CL/F) administration of YR-C-136, SR-B-103, or YR-C-108 were obtained from the NCA of the plasma data using WinNonlin (best-fit mode). The oral bioavailability (F) was also computed.

## Antiviral efficacy tests in mice for YR-C-136 and SR-B-103

Eight- to ten-week-old female BALB/c mice (Charles River Laboratories) were anesthetized with isoflurane and infected intranasally (25 μL) with a mouse-adapted SARS-CoV-2 strain[47] at 4 × $10^5$ PFU/mL in DPBS, resulting in a final inoculum of $10^4$ PFU per mouse. Inhibitor dosing (100 μL per mouse) was initiated immediately after infection by oral gavage and repeated every 12 hours. Inhibitors were pre-formulated in DMSO, polyethylene glycol 400 (PEG400), and deionized water at a 10:30:60 (v/v/v) ratio. Vehicle and Nirmatrelvir treatments were included as negative and positive controls, respectively. Mice were monitored and weighed daily until scheduled euthanasia at 48 hours post-infection. At that time, mice were euthanized and necropsied. The cranial lobe of the right lung was collected and immersed in 1 mL of DPBS in a 2-mL tube. Lung samples were weighed and homogenized using the MagNA Lyser (Roche) at 5200 × g for 1 min followed by centrifugation at 10000 × g for 5 min. The clarified supernatants were collected for plaque assay on VeroE6 cells using a previously described protocol[53]. The left lung was immersed in 10% neutral-buffered formalin (NBF) for 7 days and then processed for standard hematoxylin and eosin staining, followed by histochemistry analysis. Lung samples from uninfected healthy animals were used as normal controls. The in vivo antiviral using BALB/c female mice was performed in accordance with the guidance for the Care and Use of

Laboratory Animals of the University of Texas Medical Branch. The protocol was approved by the Institutional Animal Care and Use Committee (Protocol number 2103023) at UTMB.

### Inhibition characterizations of YR-C-136 and SR-B-103 toward SARS-CoV-2(E166V/L50F), SARS-CoV M^Pro, and MERS-CoV M^Pro

We performed the assay in the following assay buffer: 10 mM Nax-HyPO4, 10 mM NaCl, 0.5 mM EDTA, and 1.25% DMSO at pH 7.6. The final concentration of the Sub3 substrate (DABCYL-Lys-Thr- Ser-Ala-Val-Leu-Gln-Ser-Gly-Phe-Arg-Lys-Met-Glu-EDANS) and M^Pro mutant were 10 μM and 20 nM, respectively. We dissolved both inhibitors in DMSO as 10 mM stock solutions. We incubated E166V/L50F M^Pro mutant and an inhibitor in the final assay buffer for 30 min before adding the substrate to initiate the reaction catalyzed by M^Pro. The production format was monitored in a fluorescence plate reader with excitation at 336 nm and emission at 490 nm. We calculated the initial rate according to the fluorescent intensity in the first 5 min by linear regression, which was then normalized according to the initial rate of positive and negative controls. We calculated the initial rate according to the fluorescent intensity in the first 5 min by linear regression, which was then normalized according to the initial rate of positive and negative controls.

### In vitro antiviral potency tests for three SARS-CoV-2 strains and four additional coronavirus strains

The in vitro antiviral potency test (SARS-CoV-2 USA_WA1/2020 (GeneBank: MN985325); SARS-CoV-2 Delta B.1.617.2 AY.4; SARS-CoV-2 Omicron BQ.1; HCoV-Alpha 229E (GenBank: NC_002645); HCoV-Beta OC43 (GenBank: KX344031); SARS Urbani (GenBank: AY278741); MERS-CoV EMC (GenBank: NC_019843)) was conducted via the National Institute of Allergy and Infectious Diseases (NIAID) Division of Microbiology and Infectious Diseases (DMID)'s Preclinical Services. SARS-CoV-2 hCoV-19/USA/MDHP005647/2021 B.1.617.2 AY.4 (Delta) and SARS-CoV-2 hCoV-19/USA/MD-HP38960/2022 BQ1 (Omicron) were acquired from BEI Resources with catalog numbers as NR-55674 and NR-58975, respectively. Their certificates are provided in the Supplementary Information. Confluent or near-confluent cell culture monolayers of Vero E6 cells are prepared in 96-well disposable microplates the day before testing. Cells are maintained in MEM supplemented with 5% FBS. For antiviral assays, the same medium is used but with FBS reduced to 2% and supplemented with 50-μg/mL gentamicin. Compounds are dissolved in DMSO, saline or the diluent requested by the submitter. Less soluble compounds are vortexed, heated, and sonicated, and if they still do not go into solutio,n are tested as colloidal suspensions. The test compound is prepared at eight serial half-log10 concentrations. Five microwells are used per dilution: three for infected cultures and two for uninfected toxicity cultures. Controls for the experiment consist of six microwells that are infected and not treated (virus controls) and six that are untreated and uninfected (cell controls) on every plate. A known active drug is tested in parallel as a positive control drug using the same method as is applied for test compounds. The positive control is tested with every test run.

Growth media is removed from the cells, and the test compound is applied in 0.1 mL volume to wells at 2X concentration. Virus, normally at 30-100 CCID$_{50}$ (50% cell culture infectious dose) in 0.1 mL volume, is added to the wells designated for virus infection. Medium devoid of virus is placed in toxicity control wells and cell control wells. Plates are incubated at 37 °C with 5% $CO_2$ until marked CPE (>80% CPE for most virus strains) is observed in virus control wells. The plates are then stained with 0.011% neutral red for approximately two hours at 37 °C in a 5% $CO_2$ incubator. The neutral red medium is removed by complete aspiration, and the cells may be rinsed 1X with phosphate buffered solution (PBS) to remove residual dye. The PBS is completely removed, and the incorporated neutral red is eluted with 50%

Sorensen's citrate buffer/50% ethanol for at least 30 minutes. Neutral red dye penetrates living cells; thus, the more intense the red color, the larger the number of viable cells present in the wells. The dye content in each well is quantified using a spectrophotometer at 540 nm wavelength. The dye content in each set of wells is converted to a percentage of dye present in untreated control wells using a Microsoft Excel computer-based spreadsheet and normalized based on the virus control. The 50% effective (EC$_{50}$, virus-inhibitory) concentrations and 50% cytotoxic (CC$_{50}$, cell-inhibitory) concentrations are then calculated by regression analysis. The quotient of CC$_{50}$ divided by EC$_{50}$ gives the selectivity index (SI) value. Compounds showing SI values ≥10 are considered active.

Active compounds are further tested in a confirmatory assay. This assay is set up similarly to the methodology described above, where eight half-log$_{10}$ concentrations of inhibitor are tested for antiviral activity and cytotoxicity. After sufficient virus replication occurs, a sample of supernatant is taken from each infected well (three replicate wells are pooled) and tested immediately or held frozen at −80 °C for virus titer determination. After maximum CPE is observed, the viable plates are stained with neutral red dye. The incorporated dye content is quantified by regression analysis.

### Synthesis and characterization of inhibitors
All synthetic routes and characterization for the reported compounds can be found in the Supplementary Information.

### Reporting summary
Further information on research design is available in the Nature Portfolio Reporting Summary linked to this article.

### Data availability
All data are available in the main text or the supplementary materials. The crystal structures for M^Pro complexed with various inhibitors have been deposited into the Protein Data Bank with the following entry codes: 9BS7 (VP), 9BTT (SR-B-51), 9BTF (SR-B-77), 9P6F (SR-B-78), 9BTK (YR-C-108), 9BVW (SR-B-103), 9BVX (YR-C-155), and 9BSR (R-C-136). The source data underlying Figs. 4, 5, and Table 1, Table S2 are provided as a Source Data file. Viral resistance profile data access codes in NCBI SRA: SRR37076225 (wild-type SARS-CoV-2 sequence reads), SRR37076230-SRR37076233 (four biological repeats for SR-B-103 treated viruses), and SRR37076226-SRR37076229 (four biological repeats for YR-C-136 treated viruses). Source data are provided with this paper.

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

## Acknowledgements

We thank Dr. Yohannes Rezenom in the Mass Spectrometry Facility and Prof. Xin Yan of Texas A&M University for helping with the LC-MS characterizations of inhibitors. Funding: National Institutes of Health grant R35GM145351 (W.R.L.). National Institutes of Health grant R21AI164088 (S.X.). National Institutes of Health Grant R21EB032983 (W.R.L.). Welch Foundation Grant A-1715 (W.R.L.). Welch Foundation Grant A-2174 (S.X.). Texas A&M University Advancing Discovery to Market Grant (W.R.L., S.X.). Texas A&M X Grants (W.R.L., S.X.). This work was partially supported by Task Order 75N93024F00002 under Contract 75N93019D00021 from the Respiratory Diseases Branch of the National Institute of Allergy and Infectious Diseases.

## Author contributions

W.R.L., S.X., X.X., and B.W.N. conceived and supervised the research; W.R.L. and S.X. designed all inhibitors; S.A., Y.R.A., V.V., and S.N. synthesized the inhibitors and performed their purification and characterizations; K.K. conducted enzymatic $IC_{50}$ characterizations of the inhibitors; K.K. and X.S.G. performed in vitro PK analysis of selected inhibitors; D.C. and C.-C.D.C. performed characterizations of cellular $IC_{50}$ values to engage $M^{Pro}$ expressed in HEK293T cells; S.S. and C.-C.D.C. expressed and purified $M^{Pro}$ (E166V/L50F), SARS-CoV $M^{Pro}$, and MERS-CoV $M^{Pro}$; L.R.B., K.Y., and D.R. conducted the X-ray protein crystallography analysis; S.K. characterized antiviral potency of inhibitors in cell culture; H.X. and B.K.K. conducted in vivo antiviral studies and mouse tissue analysis, D.H.W. performed the histopathologic analysis and scoring; B.L.H. conducted the in vitro antiviral potency test on SARS-CoV-2 USA_WA1/2020, SARS-CoV-2 Delta B.1.617.2 AY.4, SARS-CoV-2 Omicron BQ.1, HCoV-Alpha 229E, HCoV-Beta OC43, SARS Urbani, and MERS-CoV EMC variants. All authors participated in data analysis. W.R.L., S.X., X.X., and B.W.N. drafted the manuscript with the assistance of K.K., D.C., S.A., Y.R.A., V.V., and L.R.B.

## Competing interests

The authors declare no competing interests.

## Additional information

[1]Texas A&M Drug Discovery Center and Department of Chemistry, Texas A&M University, College Station, TX, USA. [2]Department of Microbiology and Immunology, the University of Texas Medical Branch, Galveston, TX, USA. [3]Department of Pathology, the University of Texas Medical Branch, Galveston, TX, USA. [4]Department of Animal, Dairy, and Veterinary Sciences, Utah State University, Logan, UT, USA. [5]Texas A&M Global Health Research Complex of Biology and Department of Biology, College of Arts and Sciences, Texas A&M University, College Station, TX, USA. [6]Sealy Institute for Drug Discovery, the University of Texas Medical Branch, Galveston, TX, USA. [7]Department of Pharmaceutical Sciences, Irma Lerma College of Pharmacy, Texas A&M University, College Station, TX, USA. [8]Institute of Biosciences and Technology and Department of Translational Medical Sciences, College of Medicine, Texas A&M University, Houston, TX, USA. [9]Department of Biochemistry and Biophysics, College of Agriculture and Life Sciences, Texas A&M University, College Station, TX, USA. [10]Department of Cell Biology and Genetics, College of Medicine, Texas A&M University, College Station, TX, USA. [11]These authors contributed equally: Kaustav Khatua, Sandeep Atla, Demonta Coleman, Lauren R. Blankenship, Yugendar R. Alugubelli, Veerabhadra Vulupala.
✉e-mail: bneuman@tamu.edu; xuxie@utmb.edu; shiqing.xu@tamu.edu; wsliu2007@tamu.edu

