## [Peer Review file · Nature Communications]

From nicotine to SARS-CoV-2 antivirals with potent in vivo efficacy and a broad anti-coronavirus spectrum

Corresponding Author: Professor Wenshe Liu

Version 0:

Reviewer comments:

Reviewer #1

(Remarks to the Author)

This manuscript presents the discovery and optimization of novel SARS-CoV-2 main protease (Mpro) inhibitors that incorporate pyridine or its analogs as the P1 substituent. The remaining design elements were derived from known Mpro inhibitors such as nirmatrelvir. A key advantage of this compound series is their ability to bind Mpro without reliance on the critical E166 residue. Mutations at this site (e.g., E166V and E166A), which have been isolated from human patients, are particularly concerning due to their strong resistance to nirmatrelvir.

Through the integration of structural features of nirmatrelvir and 3-vinylpyridine, followed by a comprehensive medicinal chemistry campaign, the authors identified two optimized compounds, YR-C-136 and SR-B-103. Both molecules demonstrated superior in vitro potency relative to nirmatrelvir, favorable pharmacokinetic profiles, and significantly improved antiviral efficacy in virus-challenged mouse models. Notably, they retained activity against resistant Mpro variants (E166V/L50F) and exhibited broad-spectrum activity across pathogenic coronaviruses. These results strongly support the promise of these inhibitors as next-generation, standalone pan-coronavirus therapeutics and as valuable countermeasures for future coronavirus outbreaks.

The study is timely, innovative, and highly relevant to the antiviral drug discovery field. The integration of structural design, medicinal chemistry, and robust in vivo validation provides a compelling case for advancement of these molecules. Acceptance is strongly recommended.

Minor Issues to Address in Revision

1. According to the X-ray crystal structures in Figs. 2 and 3, it is the S isomer (P1 substituent) that binds to Mpro, meaning that the corresponding R isomer is not active or less active. Did the authors attempt to isolate the diastereomers to confirm their differential activity?
2. For the PK quantification in Fig. 4A and 4B, I assume the concentrations represent the total concentrations of the active isomer (S) and the inactive/less active isomer (R). Is this correct? Can they be separated in HPLC?
3. For the in vivo antiviral efficacy studies, how many days were the drugs dosed at twice/daily frequency?
4. Is the drug formulation used in the antiviral efficacy study the same as the PK studies (DMSO/PEG400/H₂O=10/30/60)?
5. The data set in Fig. 5H for YR-C-136 require explanation.
6. Were the body weight quantified in the in vivo antiviral efficacy study?
7. Other Mpro inhibitors incorporating pyridine or its analogs such as Ugi-4CR derived ML188 and its analogs can be updated in the discussion section.

Reviewer #2

(Remarks to the Author)

The manuscript by Khatuta et al. describes the synthesis and efficacy evaluation of novel Mpro inhibitors with potent broad spectrum anti-Coronaviruses activity. The presented data are indeed promising and very interesting especially that these compounds seem not be affected by the mutation at position 166 (known to confer resistance to nirmatrelvir and other Mpro inhibitors). I just have minor comments:

1. From the presented PK data in mice, the new compounds have very similar profile to nirmatrelvir, do you expect that these compounds will also need ritonavir co-administration in humans?
2. why the PK study and in vivo efficacy study were done in different type and age of mice?
3. why nirmatrelvir was not tested in parallel in the enzymatic assays in Fig 5 (A-C).
4. for Fig 5 D-J: it is not clear for me how CPE is calculated and what the values on the Y-axis means, also it would be informative to add the cell line used for each virus besides the virus names on top of each panel.
5. for the same figure 5, for panel F and H, the dose response curves are below CPE of 0.5 so EC50 cannot be calculated accurately, I suggest doing these assays starting with lower concentrations of both compounds to be able to calculate EC50.
6. Page 8 line 310: there is a reference still in the PMID format.

Reviewer #1 (Remarks to the Author): This manuscript presents the discovery and optimization of novel SARS-CoV-2 main protease (Mpro) inhibitors that incorporate pyridine or its analogs as the P1 substituent. The remaining design elements were derived from known Mpro inhibitors such as nirmatrelvir. A key advantage of this compound series is their ability to bind Mpro without reliance on the critical E166 residue. Mutations at this site (e.g., E166V and E166A), which have been isolated from human patients, are particularly concerning due to their strong resistance to nirmatrelvir. Through the integration of structural features of nirmatrelvir and 3-vinylpyridine, followed by a comprehensive medicinal chemistry campaign, the authors identified two optimized compounds, YR-C-136 and SR-B-103. Both molecules demonstrated superior in vitro potency relative to nirmatrelvir, favorable pharmacokinetic profiles, and significantly improved antiviral efficacy in virus-challenged mouse models. Notably, they retained activity against resistant Mpro variants (E166V/L50F) and exhibited broad-spectrum activity across pathogenic coronaviruses. These results strongly support the promise of these inhibitors as next-generation, standalone pan-coronavirus therapeutics and as valuable countermeasures for future coronavirus outbreaks. The study is timely, innovative, and highly relevant to the antiviral drug discovery field. The integration of structural design, medicinal chemistry, and robust in vivo validation provides a compelling case for advancement of these molecules. Acceptance is strongly recommended.

Author Response: We sincerely thank the reviewer for appreciating our work. We have included the changes requested by the reviewer in the manuscript. A point-by-point response to Reviewer 1's revision requests is given below.

Minor issues to address in revision:

1. According to the X-ray crystal structures in Figs. 2 and 3, it is the S isomer (P1 substituent) that binds to Mpro, meaning that the corresponding R isomer is not active or less active. Did the authors attempt to isolate the diastereomers to confirm their differential activity?

Author Response: We made efforts to separate the diastereomers by HPLC, and although initial resolution was achieved, the isolated fractions rapidly racemized under storage conditions. This observation suggests that the chiral centers are not configurationally stable and readily undergo interconversion, leading to the formation of a racemic mixture. The ¹H NMR spectra further supported this conclusion. We have included ¹H NMR evidence in supporting information (SR-B-103 & SR-B-122).

Pure fraction ^1H NMR of SR-B-103 in CDCl_3 after HPLC. (Slowly converting into racemic mixture)

Pure fraction ^1H NMR of SR-B-122 in CDCl_3 after HPLC. (Slowly converting into racemic mixture)

2. For the PK quantification in Fig. 4A and 4B, I assume the concentrations represent the total concentrations of the active isomer (S) and the inactive/less active isomer (R). Is this correct? Can they be separated in HPLC?

Author Response: Yes, they are total concentration. Yes, the diastereomers can be separated by HPLC, however, the isolated fractions rapidly racemized under storage conditions, preventing the maintenance of stable, enantiomerically pure species for individual biological evaluation.

3. For the *in vivo* antiviral efficacy studies, how many days were the drugs dosed at twice/daily frequency?

Author Response: We have provided the related information in the supplementary. “Inhibitor dosing (100 μ L per mouse) was initiated immediately after infection by oral gavage and repeated every 12 hours”.

4. Is the drug formulation used in the antiviral efficacy study the same as the PK studies (DMSO/PEG400/H₂O=10/30/60)?

Author Response: Yes, that is correct. We have provided this information in the supplementary. “Inhibitors were pre-formulated in DMSO, polyethylene glycol 400 (PEG400), and deionized water at a 10:30:60 (v/v/v) ratio.”.

5. The data set in Fig. 5H for YR-C-136 requires explanation.

Author Response: We thank the reviewer for pointing this out. For the HCoV Beta OC43 (H) virus, the endpoint was not determined so the EC₅₀ was listed as below the lowest test concentration (<0.032).

6. Was the body weight quantified in the *in vivo* antiviral efficacy study?

Author Response: We recorded the body weight as a standard procedure. We have provided this information in the supplementary. “Mice were monitored and weighed daily until scheduled euthanasia at 48 hours post-infection.”. However, in this replication model (at 48 h post-infection), we typically do not observe any changes in mouse body weight.

7. Other Mpro inhibitors incorporating pyridine or its analogs such as Ugi-4CR derived ML188 and its analogs can be updated in the discussion section.

Author Response: We have updated the discussion section and the reference section.

Reviewer #2 (Remarks to the Author): The manuscript by Khatua et al. describes the synthesis and efficacy evaluation of novel Mpro inhibitors with potent broad spectrum anti-Coronaviruses activity. The presented data are indeed promising and very interesting especially that these compounds seem not to be affected by the mutation at position 166 (known to confer resistance to nirmatrelvir and other Mpro inhibitors). I just have minor comments:

Author Response: We sincerely thank the reviewer for appreciating our work. We have included the changes requested by the reviewer in the manuscript. A point-by-point response to Reviewer 2's revision requests is given below.

1. From the presented PK data in mice, the new compounds have very similar profile to nirmatrelvir, do you expect that these compounds will also need ritonavir co-administration in humans?

Author Response: We thank the reviewer for the thoughtful question. Nirmatrelvir requires ritonavir boosting largely due to its rapid CYP3A4-mediated clearance and strong susceptibility to P-gp efflux, which markedly reduces its intracellular concentration (e.g., cellular IC₅₀ of 3.4 μM vs. enzymatic IC₅₀ of 66 nM).

In contrast, our lead compounds YR-C-136 and SR-B-103 show more favorable PK profiles in mice than nirmatrelvir including longer half-lives, lower clearance, higher oral exposure, and comparable or improved bioavailability (summarized in Table S1). Moreover, both compounds exhibit low susceptibility to P-gp efflux in our HEK293T cellular assay, maintaining cellular potencies much closer to their enzymatic IC₅₀ values. Together, these data suggest that our inhibitors are less likely to require ritonavir co-administration.

2. why the PK study and in vivo efficacy study were done in different type and age of mice?

Author Response: We thank the reviewer for pointing this out and we acknowledge this difference, which was due to the availability of animals.

3. why nirmatrelvir was not tested in parallel in the enzymatic assays in Fig 5 (A-C).

Author Response: Nirmatrelvir was not re-tested in our enzymatic assays because its potency values are already well established in the literature. Owen *et al.* (Science, 2021) demonstrated that nirmatrelvir potently inhibits SARS-CoV-1 M^{pro} (K_i = 4.94 nM) and MERS-CoV M^{pro} (K_i = 187 nM), establishing it as a pan-human coronavirus main protease inhibitor. More recently, Duan *et al.* (Nature, 2023) showed that the resistance-associated E166V/L50F double mutant displays a ~100-fold higher IC₅₀ compared to WT.

4. For Fig 5 D-J: it is not clear for me how CPE is calculated and what the values on the Y-axis means, also it would be informative to add the cell line used for each virus besides the virus names on top of each panel.

Author Response: Antiviral potency tests were based on inhibition of cytopathic effect (CPE) induced by different viruses. Antiviral efficacy was determined using eight half-log₁₀ concentrations of compound evaluated against a fixed concentration of virus. Cell viability was determined by neutral red uptake. Untreated and infected cells were used as a virus control and represent a CPE value of 1.0. Untreated and uninfected cell controls represent a CPE value of 0%. For each compound, a 50% effective concentration (EC₅₀) was determined using linear regression.

We have also added the cell line information for each virus besides the virus names on top of each panel of Figure 5 (D-J).

5. For the same figure 5, for panel F and H, the dose response curves are below CPE of 0.5 so EC50 cannot be calculated accurately, I suggest doing these assays starting with lower concentrations of both compounds to be able to calculate EC50.

Author Response: We thank the reviewer for pointing this out. For the Omicron isolate of SARS-CoV-2 (F) and the HCoV Beta OC43 (H) virus, the endpoint was not determined so the EC₅₀ was listed as below the lowest test concentration (<0.032).

6. Page 8 line 310: there is a reference still in the PMID format.

Author Response: We thank the reviewer for pointing this out. We have now changed this to proper reference format.